# Foodborne Toxigenic Agents Investigated in Central Italy: An Overview of a Three-Year Experience (2018–2020)

**DOI:** 10.3390/toxins14010040

**Published:** 2022-01-05

**Authors:** Valeria Russini, Carlo Corradini, Maria Laura De Marchis, Tatiana Bogdanova, Sarah Lovari, Paola De Santis, Giuseppina Migliore, Stefano Bilei, Teresa Bossù

**Affiliations:** Istituto Zooprofilattico Sperimentale del Lazio e della Toscana “M. Aleandri”—Sezione di Roma, 00178 Rome, Italy; valeria.russini-esterno@izslt.it (V.R.); carlo.corradini@izslt.it (C.C.); tatiana.bogdanova@izslt.it (T.B.); sarah.lovari@izslt.it (S.L.); paola.desantis@izslt.it (P.D.S.); giuseppina.migliore@izslt.it (G.M.); stefano.bilei@izslt.it (S.B.); teresa.bossu@izslt.it (T.B.)

**Keywords:** foodborne diseases, foodborne pathogens, epidemiological investigation, bacterial toxins, COVID-19

## Abstract

Foodborne diseases (FBDs) represent a worldwide public health issue, given their spreadability and the difficulty of tracing the sources of contamination. This report summarises the incidence of foodborne pathogens and toxins found in food, environmental and clinical samples collected in relation to diagnosed or suspected FBD cases and submitted between 2018 and 2020 to the Food Microbiology Unit of the Istituto Zooprofilattico Sperimentale del Lazio e della Toscana (IZSLT). Data collected from 70 FBD investigations were analysed: 24.3% of them started with an FBD diagnosis, whereas a further 41.4% involved clinical diagnoses based on general symptomatology. In total, 5.6% of the 340 food samples analysed were positive for the presence of a bacterial pathogen, its toxins or both. Among the positive samples, more than half involved meat-derived products. Our data reveal the probable impact of the COVID-19 pandemic on the number of FBD investigations conducted. In spite of the serious impact of FBDs on human health and the economy, the investigation of many foodborne outbreaks fails to identify the source of infection. This indicates a need for the competent authorities to continue to develop and implement a more fully integrated health network.

## 1. Introduction

Foodborne diseases (FBDs) represent a worldwide public health issue and are caused by consuming food contaminated with pathogens or their toxins. Sources of contamination are often difficult to trace and these diseases are considered easily spreadable.

Foodborne pathogens consist of bacteria, viruses, fungi and parasites contaminating food in the different phases of the production chain, during the transport, preparation and handling steps up to the final consumer. Some of these bacteria and fungi can produce toxins that outlive the producer thereof, thus the absence of the pathogen itself cannot exclude the contamination and guarantee the food wholesomeness. Many of these bacteria and their toxins are thermostable and cannot be destroyed by typical food preparation methods. As a result, the assessment of food safety becomes even more complex [1,2]. Furthermore, factors like the increase of susceptible populations (e.g., elderly and immunocompromised patients) and consumption patterns (e.g., the growing demand for the market of ready-to-eat food) have increased the risk of foodborne illnesses [3,4,5].

Bacterial toxins are classified into endotoxins and exotoxins. Endotoxins are lipopolysaccharides, originating from the outer membrane of Gram-negative bacteria after a lytic event. These molecules are thermoresistant, moderately toxic and can act locally [6]. On the other end, exotoxins are heat-labile proteins produced by both Gram-negative and Gram-positive bacteria through lysis or secretion. These molecules are more specific and can be effective at low doses and at a distance from the production site. In turn, exotoxins can be divided into neurotoxins, enterotoxins and cytotoxins according to the physiological effect they cause [1,2]. Bacterial exotoxins are mainly produced by *Bacillus cereus*, *Clostridium botulinum*, *Clostridium perfringens* and *Staphylococcus aureus*.

The most commonly used detection and quantification methods for those toxins are: biological assays (whole animal and cell culture assays), immunological assays (enzyme-linked immunosorbent assays—ELISA), reversed passive latex agglutination assay (RPLA) and lateral flow immunoassay (LFIA), mass spectrometry, as electrospray ionization (ESI) and matrix-assisted laser desorption/ionization assays (MALDI). Furthermore, molecular assays, such as PCR and whole genome sequencing, can be used for virulence genes detection despite the fact that a pathogen carrying the toxin-encoding genes could not be able to express the toxin [7]. The gold standard for the detection and biological characterisation for some toxins are still animal bioassays [7]. 

The other main bacterial pathogens responsible for FBDs are *Campylobacter* spp., *Salmonella* spp., *Listeria monocytogenes*, Shiga toxin-producing *Escherichia coli* (STEC) and *Yersinia* spp. In addition to them, some viruses are responsible for FBDs as well, mainly *Norovirus* and *Hepatitis A virus* (HAV) and *Hepatitis E virus* (HEV). The identification of these pathogens from food and the environment is mainly based on the use of cultural and molecular methods [8]. Further subtyping and characterisation are achieved through serological, enzymatic and PCR-based methods, Sanger sequencing and next-generation sequencing [8,9].

According to the Directive 2003/99/EC, member states of the European Union must collect data on the occurrence of zoonoses, zoonotic agents, antimicrobial resistance and foodborne outbreaks. Data are examined by EFSA and published in the annual European Union Summary Reports, in cooperation with the European Centre for Disease Prevention and Control (ECDC). In particular, reporting of cases of infection from *Salmonella*, *Campylobacter*, *Listeria monocytogenes*, Shiga toxin-producing *E. coli* (STEC) as well as information on foodborne and waterborne outbreaks (FBOs) are mandatory while hepatitis A, botulism and yersiniosis must be reported based on the actual epidemiological situations [10].

In the last report of EFSA and ECDC on human zoonosis, the epidemiological situation of the principal foodborne pathogens responsible for FBDs in Europe during 2019 has been outlined [10]. Campylobacteriosis and salmonellosis were the most commonly reported gastrointestinal infections in humans in the EU. Salmonellosis is the first cause of foodborne outbreaks in the EU/EE (17.9%), the majority of which is caused by *S.* Enteritidis. Listeriosis, caused by *L. monocytogenes,* is one of the most serious foodborne diseases under EU surveillance. *Listeria* foodborne outbreaks are not common, but in 2019 they showed the highest percentage of death (8.9%), followed by botulinum toxin (5.9%). Yersiniosis was the fourth most commonly reported zoonosis in humans in 2018 and 2019. Yersiniosis cases in humans and outbreaks caused by *Yersinia enterocolitica* were stable and flat in 2015–2019. 

Overall, identified or unspecified bacterial toxins were the first cause of all foodborne outbreaks reported in 2019 in the EU (19%). The bacterial toxins identified as produced by *B. cereus*, *C. perfringens*, *S. aureus* and *C. botulinum* caused about 6% of the foodborne outbreaks.

Shiga toxin-producing *E. coli* was the sixth most frequent foodborne outbreak bacterial agent detected in the EU, and the trend has been increasing from 2015 to 2019, with the highest notification rates reported in Ireland, Malta, Denmark and Sweden. 

In 2019, *Norovirus* (and other Calicivirus) was the second most frequently reported causative agent in FBOs in the EU, being associated with large outbreaks (24.3 cases on average).

Only five EU member states reported data related to HAV or other unspecified Hepatitis viruses, with a total of 135 cases in 2019. Compared with 2018, the number of notified Hepatitis A (including other Hepatitis virus, unspecified) decreased in the EU, mainly due to reduced reporting. Hepatitis A outbreaks were characterised by a high percentage of cases needing hospitalisation (73.3%).

The growing public health focus on microbial toxins and bacterial agents is due to a set of determinants including a better overall surveillance and an increase in the number of notified foodborne outbreaks, including those involving bacterial toxins [7].

The reported number of zoonosis outbreaks to EFSA and ECDC in Italy in 2018 and 2019 were the same (134 and 135, respectively) [10,11]. Particularly, in 2018 the 25% of outbreaks were caused by *Salmonella* spp., the 13.3% by bacterial toxins, the 11.2% by *Campylobacter* spp. and the 0.7% by Shiga-toxins producer *E. coli* (STEC). In 2019, *Salmonella* spp. outbreaks decreased to 13.3% of overall outbreaks, bacterial toxins maintained the same levels while *E. coli* (STEC) rose to 1.5%. *S. aureus* and *C. perfringens* caused the same number of outbreaks in 2018 and 2019, *C. botulinum* outbreaks decreased from 4.5% to 1.5% and *B. cereus* slightly grew (from 0.7 to 3%).

The number of outbreaks caused by HAV was stable for the two considered years (3%). In 2019, cases from other Hepatitis virus were also reported (1.5%). In contrast to the European trend, the percentage of Italian outbreaks caused by *Norovirus* was lower, and decreased from 6.7% in 2018 to 4.4% in 2019. In 2019, only one outbreak related to *L. monocytogenes* was reported, involving 12 cases and causing two deaths. No yersiniosis-related outbreaks were notified from Italy in 2018 and 2019.

Multi-level monitoring, including contamination in the food chain control subsequent steps from distribution to consumers, human disease surveillance and epidemiological investigations of epidemics and sporadic cases, is still an important source of information for authorities to assess the success of current food safety management systems and to identify new hazards [12]. 

Outbreak surveillance primarily aims to stop the outbreak by identifying the offending products and withdrawing them from the market. Investigations can aim to identify all the involved cases and find out the unsafe practices that led to the outbreak [12].

Furthermore, most pathogens that can be transmitted by food, may also be transmitted by other pathways such as water, direct human and animal contact. Therefore, there is a need for source attribution to quantify the proportion of all the foodborne cases, and the food vehicles that are most frequently associated with illness [12,13].

Italy entrusts scientific and institutional research on FBDs to the Istituto Superiore di Sanità (ISS) which coordinates surveillance of FBDs at the national level, through the Department of Food Safety, Nutrition and Veterinary Public Health and the Department of Infectious Diseases. They host numerous National Contact Points designed by the Ministry of Health for the collection of surveillance data. The department includes different laboratories and reference centres for many microbiological agents of disease. Other activities on specific pathogens or pathologies are distributed between National Reference Centres (NRCs) and National Reference Laboratories (NRLs). The CRNs represent an operational tool of high and proven competence at the service of the state, in the sectors of animal health, food hygiene and zootechnical hygiene. They are located in different Experimental Zooprophylactic Institutes (IZSs) and identified by the Ministry of Health through ministerial decrees. NRLs are established according to the Reg. (EU) 2017/625 both in IZSs and in ISS and sometimes both NRCs and NRLs are set up at the same laboratory for a specific activity. Concerning FBDs’ related pathogens and their characterisation, the following organisations are currently active in Italy: NRC and NRL for Salmonellosis (IZS delle Venezie, IZSVe), NRC for Emerging Risks in Food Safety (IZS della Lombardia e dell’Emilia Romagna, IZSLER), NRC for Chemical and Microbiological Control in Bivalve Mollusks (IZSVe), NRC and NRL for Antimicrobial Resistance (IZS del Lazio e della Toscana, IZSLT), NRC for Whole Genome Sequencing of Microbial Pathogens: Database and Bioinformatics Analysis (GENPAT) (IZS dell’Abruzzo e del Molise, IZSAM), NRL for *L. monocytogenes* (IZSAM), NRL for *Campylobacter* (IZSAM), NRL for Coagulase Positive Staphylococci and *S. aureus* (IZS del Piemonte Liguria e Valle d’Aosta, IZSPLV), NRL and EURL for *E. coli* at Italian National Institute of Health (ISS), National Reference Centre for Botulism (ISS) and NRL for Foodborne Viruses (ISS). These organisations act as a reference in the conduct of epidemiological investigations, especially when they provide periodic reports and manage platforms on which positive samples are loaded with the relative metadata and the results of the typing tests. For pathogens with no reference laboratories assigned, such as toxin-producing *B. cereus* and *C. perfringens*, obtaining surveillance data and samples to compare with becomes more complicated. 

In Italy, notification of infectious disease is regulated by law DM 15 December 1990, that categorises notifiable diseases into five classes according to their importance and impact on public health. Infectious and FBDs pathogens and bacterial toxins belong to classes I, II and IV: -botulism is included in class I diseases for which an immediate report is required (within 12 h);-listeriosis, hepatitis and non-typhoid salmonellosis are 2nd class diseases because of their high frequency and require a report within 48 h from the observation of the case (also for suspected diseases);-other FBDs are included in the 4th class. In this case, the individual medical report must be followed by reporting from the local health unit only if an epidemic outbreak occurs.

Data are reported to the Ministry of Health Directorate General Health Prevention Transmissible Diseases and International Prophylaxis Unit. The notification flow is distinct for sporadic cases (class II) and outbreaks (class IV). Human cases of botulism and trichinellosis are subject to mandatory notification within 12 h.

With the exception of the above-mentioned notification system, no national protocol of intervention for the management of foodborne outbreaks is available; therefore, the management of this system is actually entrusted to the individual regional authorities. 

As a consequence, the system suffers from extreme fragmentation and lack of standardisation, with the establishment of a variable number (0, 1 or 2) per region of Regional Reference Laboratories (only nine of which are formally recognised) for the management of FBDs and related operational guidelines (not always available). Furthermore, other public institutions occasionally collect the same kind of data in the same area and the lack of shared database renders an even more partial the view on the toxic infections currently occurring in each region.

Our laboratory, historically operating as an official control laboratory for food microbiological analyses, was also recognised as Regional Centre for Enteropathogenic Bacteria in 1996 and afterwards was appointed as actor in the Regional Plan for the Surveillance and Management of Infectious Emergencies during the Extraordinary Jubilee 2015–2016, concerned the surveillance and notification of foodborne disease, with particular regard to *Salmonella* from human, animal and environmental sources. It collects bacterial isolates obtained through the microbiology control of food matrices sampled by competent authorities of Lazio and Tuscany regions and from the producers themselves for self-control analyses. Furthermore, it receives human isolates from public and private hospitals and laboratories across the entire area of competence for serological typization. In order to overcome the above-mentioned critical points, the Lazio region has been recently appointed IZSLT as the Regional Reference Laboratory for FBDs and foodborne human diseases (Deliberation of Lazio Region, n. G06447 of 28 May 2021). The deliberation determined the organisation of a regional group for the management of FBDs, which will lay down guidance for the surveillance of FBDs. The regional deliberation should end the overlapping of activities and promote the centralisation of epidemiological and analytical data, which must then be communicated to the Regional Service of the Monitoring of Infectious Diseases at INMI L. Spallanzani (SERESMI). 

In this study, we collected the experience of the Istituto Zooprofilattico Sperimentale del Lazio e della Toscana “M. Aleandri” (IZSLT) in identifying pathogens and bacterial toxins in food and the environment, prior to the official appointment of our laboratory. The investigated samples were collected by the competent authorities, mainly in the Lazio and Tuscany regions, in relation to all the FBD reports notified to our laboratory for the three-year period 2018–2020.

## 2. Results

We collected data related to 70 investigations for FBDs: 28 cases in 2018, 29 cases in 2019 and 13 in 2020. Only 17 out of 70 investigations (24.3%) took place following an official FBD notification by the health regional or national system (hospital, emergency room ER, or primary care physician) with the patient having a clinical diagnosis of FBD. Among the remaining cases, 29 (41.4%) investigations started with a direct consumer report after a symptomatic event (e.g., gastrointestinal symptoms, fever, headache, etc.) with or without a clinical diagnosis, and 24 (34.3%) followed a notification from the competent authorities with no information on the patient’s condition provided to our laboratory during the investigation (Appendix A).

The overall number of undertaken investigations were similar for 2018 and 2019, but dropped drastically in 2020 (Figure 1). Overall, 70% of the investigations were conducted both in small and large food retailers and catering services (school, company and hospital canteens, and restaurants) and 28.6% in private homes (Table 1 and Appendix A). A total of 340 samples were analysed. For each investigated case, a variable number of samples was collected (ranging between 1–67 samples, mean 4.84, median 2). In total, 49.1% of the food matrices represented the “Ready-to-eat” (RTE) category, 47.6% were non-RTE, with the remaining 3.2% obtained from the immediate environment (sponge-wiped surfaces).

Nineteen (5.6%) out of 340 samples analysed and which involved 17 investigations, were positive for the presence of food-related pathogens. Of these, twelve (63.1%) represented non-RTE foods, while five (26.3%) represented RTE foods. Among the positive samples, more than half (11) represented meat-derived products (five pork, four poultry and two bovine), of which only three samples were RTE foods. 

In total, 36.5% of the food samples analysed represented leftovers of meals consumed by the patient with evidence from the available data (6.3% positives); 14.1% were meal leftovers probably consumed from patients without assessed evidence (4.2% positives); 20.3% were sampled from the same package or batch of the suspected meal (no positives found); 22.9% were samples collected from the market retailers and restaurants pointed to by the patients (9% of positives); 2.9% were unrelated samples collected still sealed in patients’ private homes (no positives found); and 3.2% were environmental samples (18.2% of positives) (Table 2). 

The time ranging between the onset of symptoms and sample collection by the competent authority was measured for 28 investigations, with a mean value of 8.3 days (min 1, max 26 days).

The commonest pathogen detected was *L. monocytogenes* (six of 19 positives), of which two represented environmental samples corresponding to investigation number (IN) 58 and IN70 of Appendix A, two were from non-RTE meat products (IN53, IN63), one was an RTE meat product (IN58), and one was a non-RTE vegetable product (IN59) (Table 2); of the six positive, five were reported in 2020. One RTE poultry sample was positive for *Salmonella* Infantis (IN44), while a non-RTE bovine meat sample was positive for *E. coli* (STEC) (IN34). One case of *Y. enterocolitica* was identified in an RTE pre-cooked pasta (IN39). Two cooked homemade preparations were positive for *Clostridium perfringens* (IN28, IN51); in one of these, the food was also *Bacillus cereus* contaminated. Coagulase-positive *Staphylococcus* was detected twice, once in RTE fresh ovine cheese (IN14) and once in non-RTE chicken (IN66). Staphylococcal enterotoxins were detected in two cooked dishes prepared by catering services at two separate localities (IN23, IN26). With regard to viral pathogens, *Norovirus* and HAV were detected once in the same samples of fresh mussels (IN7), while HEV was detected twice in non-RTE pork sausages from one producer and linked back to three affected patients (IN42). In one case, the RTE roasted pork was infested with larvae of the fly genus *Lucilia* (Calliphoridae) (IN65). In total, 36.8% of the positive samples presented foodborne toxigenic agents.

Three diagnosed cases of botulism were notified. The first (IN25) involved a hospitalised patient, who in the days prior to his illness had consumed locally peddled food. Several foods, including cheese, bovine hamburgers, pork salamis and vegetables in oil, were sampled but none were *Clostridium botulinum* positive. Another case (IN68) concerned the recall of a particular batch of canned tuna in sunflower oil, originating in an outbreak of botulism amongst 37 people on the island of Sicily who had all visited the same canteen. During this investigation, none of the 67 cans of tuna analysed were positive for *C. botulinum* toxins. The recall was accordingly revoked. The third case involved one person being hospitalised (IN69). All of the food samples analysed, including the homemade tuna kept in oil, tested negative for any *C. botulinum* toxins.

In two HAV investigations (IN4, IN7), only one was linked to contaminated farmed mussels (IN7), and in two cases of HEV (IN3, IN42), only one was probably caused by the contamination of both fresh and seasoned pork products sampled in a butchery shop (IN42). Regarding the cases of diagnosed salmonellosis (IN19, IN32, IN56), the investigations did not give any result in all three of them. 

Of the seven cases of listeriosis investigated, only three yielded samples that were positive. In these cases, all occurring in 2020, the pathogen was isolated both from the patient and the suspected food, and additionally typed with NGS (whole genome sequencing, WGS) to compare the strains and perform a deep epidemiological investigation. In the first case (IN63), this molecular approach allowed us to exclude any link between the strain isolated from the patient and the strain isolated from fresh minced beef sampled in the patient’s home. The origin of patient contamination remains unknown to date. The second case (IN58) involved a pregnant woman and, after investigation at the frequented market, a pork RTE meat was found positive to the same serotype and ST. This case is still under study for evaluating the relatedness by comparing the genomic data. In a third case (IN70), a nosocomial outbreak of listeriosis was detected in the Lazio region. The source of contamination was attributed to the meat slicer of the hospital kitchen by using a WGS approach as reported by Russini et al. [14].

When a specific clinical diagnosis was not available (53 out of 70 investigations), the search for the responsible pathogens focused on the most common cause of FBD, according to the characteristics of the food matrix involved (Appendix A). The principal investigated pathogenic agents were the enterotoxins producer bacteria coagulase-positive *Staphylococcus*, *B. cereus*, the staphylococcal enterotoxins, *Salmonella* spp. and *L. monocytogenes*. In 12 cases, we found a positivity in the collected food sample, and the most frequent identified targets were staphylococcal enterotoxins (2), positive coagulase *Staphylococcus* (2), *C. perfringens* (2) and *L. monocytogenes* (2) (Appendix A). In eight cases, the positive matrix was a meat sample (belonging both to the RTE and non-RTE category).

In one important investigation, which occurred in a police school canteen (IN38), 130 persons were involved, and no specific diagnosis was formulated for all the symptomatic people. Served meal samples and environmental samples were collected. The analyses tested a wide range of pathogens but no positive samples were found (Appendix A).

## 3. Discussion

For several years, the need for studies concerning FBDs has been globally recognised. Estimating the burden of FBDs is necessary in order to reach a global risk ranking for policy making. For this purpose, the harmonisation of laboratory methodologies, epidemiological and biological data collection and sharing are useful processes for comparing the estimates between diseases, countries and regions [15]. 

This report describes the experience of a bi-regional (Lazio and Tuscany) focus on the analysis for FBD investigations, which occasionally received cases that occurred in other Italian regions, in the three-year period (from 2018 to 2020).

Considering this time frame, we noticed a general decline in cases during 2020, with 13 investigations, compared to 28–29 in the previous years. Particularly, 2020 was characterised by a collapse of investigations carried out in restaurants, cafeterias and holiday dinners both for the number of food sampling sites and for the number of food sampling preparation sites. In parallel, food samples prepared or consumed at home increased. Given the peculiar situation due to the COVID-19 pandemic during the whole of 2020, consumers’ habits have inevitably changed [16]. The harsh lockdown restrictions caused the closure of restaurants, schools and company canteens, probably leading to a decrease in cases involving this kind of service. As a matter of fact, a preliminary study performed in Spain showed a marked decrease in the number of reported foodborne infections during the first semester of 2020, compared with the same of 2019, specifically for *Campylobacter* and *Salmonella* infections [17]. Accordingly, during the first Italian lockdown (February–May 2020) we did not receive any reports or samples linked to a suspected or overt case of FBD. 

A probable collateral effect of the pandemic could be the avoidance of emergency rooms or medical care in the case of FBDs without acute symptoms. The reduction of tourist flows may also have played an important role in reducing infectious diseases, which collapsed during the first months of lockdown [17].

The impact of the COVID-19 epidemic on public health is not limited to this aspect. Many countries reported that in 2020, and specifically during the first half of the year, less non-COVID-19 patients were hospitalised. In a large University-Hospital in Parma, Italy, admissions for non-communicable diseases (NCDs) in 2020 vs. 2019 dropped by approximately one third [18]. Denmark reported a significant decrease in hospital admission during national lockdowns compared with the pre-pandemic baseline period [19]. According to a survey carried out in the U.S. on healthcare staff, several routinary services, including investigations related to other communicable diseases, foodborne outbreaks, public health surveillance and evaluation, and non-communicable disease responses were no longer available or heavily penalised, due to the burden of the COVID-19 impact. Foodborne outbreaks were specifically mentioned, highlighting a reduced ability to conduct surveillance, outbreak investigation or inspections [20]. To our knowledge, similar reports concerning the COVID-19 effects on FBDs in Italy have not been disseminated, but we can hypothesise that the drop in notifications may have had the same cause.

Even before the COVID-19 pandemic, a general underestimate of the population burden of FBDs was suspected by the official bodies; in most countries, illness recording occurs only when a patient consults a doctor or a nurse, and they require a sample for laboratory testing [21]. Only for severe cases, where hospitalisation is needed, it proceeds with the clinical investigation of the causes and the detection of the pathogens and toxins involved. In addition, where the symptoms have been ignored or confused, the customers’ complaints are infrequent.

In the U.S., only about half of the foodborne illness outbreaks have a recorded contributing agent, and data describing the outbreaks are limited and not sufficient for further analysis [22]. A retrospective telephone survey conducted in Italy between 2008 and 2009 reports that only 39.5% of persons with a self-reported episode of gastrointestinal illness contacted a physician; only 0.3% of the total submitted a specimen for laboratory investigation [23].

Another consequence of the scarce sensibility of the surveillance systems is the difficulty to detect the decrease of the occurrence of a specific foodborne illness due to the success of control programs against the pathogen responsible for the disease. This fact increases its relevance when the numbers of the observed illnesses are small or underestimated [24]. Retrospective telephone surveys have been proposed to be a cost effective tool to detect changing disease incidence, but they cannot estimate the contribution of specific pathogens [25]. Expert elicitation, on the other hand, can be used to determine exposure routes for key hazards such as foodborne bacteria, but they can suffer from substantial uncertainty [26].

Out of a total of 340 analysed samples, only 19 were positive for the presence of a foodborne pathogen with 26% belonging to the RTE category. The reasons for failing in positive detection may be different. From a clinical point of view, the impossibility of carrying out a correct source attribution could be due to the lack of clinical indications that could direct the selection of pathogens to be searched. In our experience, only 24.3% of investigations started with a diagnosis of FBD and, with the exception of botulism, no clinical diagnosis related to toxin agents was found. In general, these kinds of diseases are not deeply investigated due to the symptoms that are often generic or treated with a symptomatological clinical approach, from both the patients and the medical health system (e.g., emergency room). As a consequence, an inaccurate anamnesis and diagnosis may not be able to correctly address the search for pathogens that could thus escape.

A robust detection of pathogens is also favoured when the period of time elapsed between the moment of possible intoxication and the sampling by the official authorities is short. In this case, it could be easier for the patient to remember the meals consumed and correctly address the epidemiological investigation on the most probable sources of contamination. In our case series, the average period was 8.3 days, with a maximum of 26 days. This value is given by the incubation period (which is the time from infection with the pathogen to the onset of symptoms), the time occurring from the onset of symptoms to the consultation of a physician or a competent authority and those necessary to organise the sampling procedures after notification of the event. It is reasonable to think that in many cases it was impossible to sample a residue of the suspected meal, since it is not admissible to sample food after its expiration date (in many cases, packaged food is considered expired after a few days after its opening), preventing the investigation on the most probable source of the pathogen. Other complications are given by the unavailability of the contaminated food since it was completely consumed or sold. These limits are often overcome by sampling food from another package from the same batch (20.3% of our case series with no positivity found), or from samples collected from the market retails and restaurants pointed to by patients (22.9% of our case series with a 9% positivity rate), making uncertain any assumption on the origin of the involved pathogen.

A technical issue to consider is related to some intrinsic limits of sampling protocols and detection procedures. Given the uneven distribution of biological contaminants in food, the chance of detection may depend on the type of sampling (e.g., if the food sample is composed of more than one increment or if the test portion is unique) [27]. During the evaluated years, in two positive cases, the parameters analysed did not exceed the legal limits. Therefore, it could be possible that the involved patients have been exposed to portions of the meal more contaminated than the portion sampled by the authorities during the investigation.

When clinical indications and anamnesis are absent, food and residual samples are subjected to a vast panel of tests, and it may happen that the quantities were lower than that indicated by the reference standards for certain microbiological targets. If the food sample is not available or available in limited quantities, based on the investigated pathogen, it would be advisable to carry out environmental sampling, but this type of procedure is almost never carried out (we could examine only 11 environmental samples related to nine out of 70 investigations).

According to the definition of “Ready-to-eat” food (RTE) of commission regulation (EC) No 2073/2005 as food intended for direct human consumption, numerous factors may enhance the hazard level of RTE. Surfaces may be reservoirs for bacterial contamination, which could increase the risk of bacterial aggregation and dissemination during RTE preparations and manipulations (e.g., slicing or packaging), exposing the final consumer to foodborne pathogens. This hygienic issue is worsened by the ability of some bacteria to form biofilm, which facilitates their survival on surfaces and protects them from drying and cleaning procedures [14,28]. RTE foods prepared by hand are often implicated in foodborne illness outbreaks, as this direct contact may lead to an increased incidence of contamination with potential foodborne pathogens [28,29].

In our report, 63.1% of positive samples belonged to the category of non-RTE food and 26.3% were RTE food. Despite the majority of samples not being considered RTE, 52.6% of positive samples were residual or probable residual of consumed food (two RTE and eight non-RTE samples). Of the eight non-RTE foods, seven were prepared with fresh meat (beef, pork and chicken) and one with a soft cheese and all were manipulated in private houses or canteens. In addition to the already mentioned factors that may increase the hazard in the production of RTE food, the risk of FBDs appears to increase in fresh food not properly cooked or handled after cooking. Therefore, the incorrect treatment, handling and consumption of food of any category (not necessarily RTE) seems to be a crucial aspect that should not be underestimated. Collecting environmental samples, particularly for some pathogens, could be crucial for detecting the role of secondary contamination in FBDs [30,31].

The NGS approach is currently an irreplaceable and increasingly used tool in the study of outbreaks, in epidemiological investigations and source attribution studies [9,32,33]. This approach could also help in the genomic characterisation of toxigenic pathogen strains, and evaluate the presence of genes associated with toxins production, assess the genetic variants and monitor the spread of antimicrobial resistance [34,35,36]. We applied this methodology only when both the isolates of food and human origin were available (three cases of listeriosis). In one case, the analyses carried out gave inconsistent results since the strain of *L. monocytogenes* isolated from food found in the patient’s house was not compatible with the one isolated from the patient. In another case, the NGS approach was decisive in tracing the source of contamination [14].

This aspect underlines the usefulness of simultaneously managing both strains of animal, food and environmental origin (deriving from the official control activities of our laboratory) and human origin (collected by virtue of the regional acknowledgments obtained over time), according to a One-Health perspective. Furthermore, this work highlights the central importance of the collaboration of the different institutions (IZS, ISS, regional and local competent authorities that take on the surveillance of FBDs), even if further interventions for the standardisation of procedures and intensification of data exchange networks are required. In this context, our report could provide a further dissemination channel, representing a source of analytical information useful for more geolocated study on FBDs in addition to official transmission routes that expose data in a general and aggregate manner.

This study brings to attention, even if secondary to the main objective, the critical issues related to the study and research of FBDs in the era of COVID-19, suggesting how the pandemic could also have affected the frequency and identification of these diseases.

In conclusion, our report could contribute to estimates of the burden of FBDs, to harmonise methodologies and to share data. Our experience, even if conducted with few exceptions at a regional level, can highlight some critical and operational aspects (such as the regulatory gap, absence of standard guidelines for the management and organisation of interventions during epidemiological investigations) that can also be generalised to wider levels of FBD management.

## 4. Conclusions

The official data regarding notifications of cases and outbreak of FBDs in Italy may suffer from a general underestimation compared to European trends. The present work reports the result of three years of investigations related to the episodes of FBDs reported to the Food Microbiology Unit of the IZSLT, before the official appointment as a Regional Reference Laboratory. The highlighted criticalities were linked to a fractional management of FBDs, which still characterises various regions of our territory, to the lack of integrated coordination at the national level as well as to a disparity of treatment and knowledge with respect to the different foodborne pathogens. Therefore, the same issues could be relevant in other Italian regions.

From a technical point of view, it is necessary to implement the use of certain methods with a high discriminating power (i.e., NGS-based) and to develop integrated management platforms for sharing metadata and analytical data. In conclusion, a better assessment of FBD outbreaks can give better information to the risk managers, leaving more space for more accurate monitoring of the impact of the implementation of their decisions.

## 5. Materials and Methods

### 5.1. Epidemiological and Clinical Data Collection

Data concerning the 70 FBDs investigations were collected for the three-year period 2018–2020, during the routine laboratory activities of food control of our institute, covering Lazio and Tuscany regions, which occasionally received cases that occurred in other Italian regions. The main sources of information were the official report of sampling issued by the local competent authority and, when available, food questionnaires administered to patients during hospitalisation were a further source of data collection and the starting point for investigations. The records considered for this study include both investigations started by the competent authority after the notification of an FBD (listeriosis, salmonellosis, etc.) by the involved hospital who treated the patient and isolated the pathogens, or after personal complaints from consumers who declared they had symptoms after consuming food in hospitals, emergency rooms or at home, without a clinical diagnosis.

### 5.2. Microbiological and Molecular Analyses

The local competent authorities carried out environmental and food sampling, the latter consisting of remains of meals in private homes and restaurants, or foodstuff from the same production batch sold in the supermarkets frequented by the patients. If the suspected batch of food was no longer available (fully consumed, expired or withdrawn), the authority proceeded with sampling a different batch of the same product or a similar product produced by the same company.

Detection and identification of pathogens and toxins from food and environmental samples were performed by the Food Microbiology Unit of IZSLT through internal procedures, proprietary protocols and the standard tests defined by the international and European standard described in Table 3. When required, both molecular detection and cultural microbiological methods were performed.

The bacterial isolates of human origin were obtained by the Microbiology Unit of hospitals involved in the investigations and transferred to the Regional Reference Centre for Pathogenic Enterobacteria (CREP) at the Food Microbiology Unit of IZSLT for serological and molecular typing.

### 5.3. Data Analyses

The samples were first divided by the collection points, classified into three groups: “private house”, concerning the food consumed at home or collected inside of patients’ or consumers’ home; “restaurant”, including restaurants, catering services, hospital kitchens, workplaces and school canteens; “retail store”, concerning all type of retail, e.g., local outdoor market, food shop, butchery and supermarket. Subsequently, the foodstuffs were divided into two subgroups depending on the place of production: prepared meals manipulated at home (cooking or homemade) or in the restaurant, and those purchased in retail stores and not modified or further processed. The food directly sampled from retail stores and public services were not divided in subgroups.

For the classification of food as Ready-to-eat (RTE) or not (non-RTE), we followed the Commission Regulation (EC) No 2073/2005, defining RTE food as “food intended by the producer or the manufacturer for direct human consumption without the need for cooking or other processing effective to eliminate or reduce to an acceptable level microorganisms of concern”. We used as standard the list of RTE food categories identified by the European Food Security Agency (EFSA) [10]. 

## Figures and Tables

**Figure 1 toxins-14-00040-f001:**
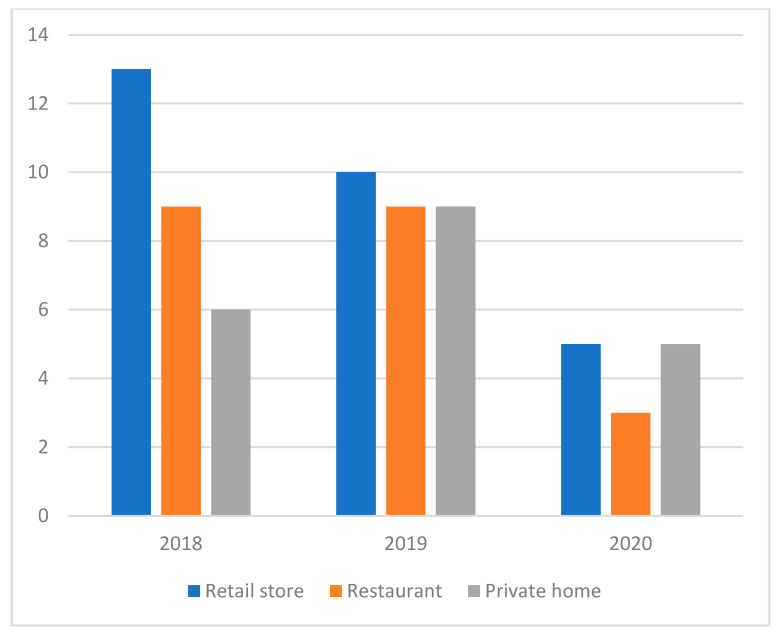
Annual breakdown of inspections conducted at various locations situated in the Tuscany and Lazio regions of Italy (2018–2020). Inspections required by customs services were omitted.

**Table 1 toxins-14-00040-t001:** Enumeration of samples according to place of sampling, origin of foodstuffs and positivity assessment according to the panel of the performed analyses.

Sampling Source	Foodstuff Origin	Analytical Result	No.
Private home	Homemade	Positive	2
		Negative	53
	Retail store	Positive	4
		Not executable	1
		Negative	44
Restaurant	Restaurant	Positive	6
		Negative	102
	Retail store	Positive	0
		Negative	4
Customs services	-	Positive	0
		Negative	2
Retail store	-	Positive	7
		Negative	115
Total samples			340

**Table 2 toxins-14-00040-t002:** Number of samples and percentage of positives per matrix.

Matrices	Negative	Positive (%)	Positivity for Pathogens or Toxins	Total	% RTE
Environmental samples	9	2 (18.18)	*Listeria monocytogenes* (2)	11	-
Meat	64	11 (14.67)	*E. coli* STEC (1) ***Clostridium perfringens*** (1) **Coagulase Positive *Staphylococcus*** (1) *Listeria monocytogenes* (3) **Staphylococcal enterotoxins** (1) Fly larvae (1) HEV (2) *Salmonella* Infantis (1)	75	45.33
Products thereof (fish, sauces, pasta)	117	3 (2.5)	***Clostridium perfringens*** (1) ** ***Bacillus cereus*** (1) ** **Staphylococcal enterotoxins** (1) *Yersinia enterocolitica* (1)	120	75.21
Milk and milk products	33	1 (2.94)	**Coagulase Positive *Staphylococcus*** (1)	34	88.24
Honey and pastry	6	0 (0)	-	6	40
Fish	21	1 (4.34)	*Norovirus* (1) ** HAV (1) **	23 *	26.09
Eggs	9	0 (0)	-	9	0
Vegetables	61	1 (1.61)	*Listeria monocytogenes* (1)	62	9.84
Total samples	320	19		340	49.1

Samples positive for toxins or corresponding toxin-producing bacteria highlighted in bold. * One sample not suitable for analysis. ** Dual contamination.

**Table 3 toxins-14-00040-t003:** Protocols and procedures used to detect the targets and their reference.

Target	Technique	Reference Standard
*Bacillus cereus*	Cultural examination—fcu count	ISO 7932
*Campylobacter* spp.	Cultural examination—detection	ISO 10272-1
*Campylobacter* spp.	PCR Real Time—detection	iQ-Check^®^ Campylobacter AOAC 031209
Toxin-producer *Clostridium botulinum*	Cultural examination—detection	ISS CNRB30
*Clostridium perfringens*	Cultural examination—fcu count	ISO 7937
*Clostridium perfringens* enterotoxin gene (CPE)	PCR multiplex—detection	Internal procedure
Enterobacteriaceae	Cultural examination—fcu count	ISO 21528-2
Staphylococcal enterotoxin	ELFA—detection	ISO 19020
*Escherichia coli*	Cultural examination—fcu count	ISO 16649-2
*Escherichia coli* STEC	PCR Real Time	ISO/TS 13136
AIL gene	PCR Real Time	ISO/TS 18867
Botulinum toxin genes	PCR Real Time—detection	ISS CNRB31
*Norovirus* G1 and G2	PCR Real Time reverse transcription—detection	ISO/TS 15216-2
*Listeria monocytogenes*	Cultural examination—detection	ISO 11290-1
*Listeria monocytogenes*	Cultural examination—fcu count	ISO 11290-2
*Listeria monocytogenes*	PCR Real Time—detection	iQ-Check^®^ Listeria monocytogenes AFNOR BRD 07/10-04/05
*Listeria monocytogenes*	NGS—whole genome sequencing	Nextera XT DNA Library Preparation Kit Illumina MiSeq System
Potentially enteropathogenic vibrions	Cultural examination—detection, PCR Real Time, PCR end point	ISO 21872-1
*Salmonella* spp.	Cultural examination—detection	ISO 6579-1
*Salmonella* spp. serovar	Serum agglutination	ISO/TR 6579-3
*Salmonella* spp.	PCR Real Time—detection	iQ-Check^®^ Salmonella AFNOR BRD 07/06-07/04
*Shigella* spp.	Cultural examination—detection	ISO 21567
Coagulase-positive staphylococci	Cultural examination—count fcu	ISO 6888-2
Staphylococcal enterotoxin type A, B, C, D, E, G, H, I, J, P, R	Multiplex PCR Real Time—detection	SOP IZSPLV N. 10CA186
*Botulinum* toxins	Mouse test	ISS CNRB30
*Hepatitis E virus* (HEV)	PCR Real Time reverse transcription—detection	Internal procedure
*Hepatitis A virus* (HAV)	PCR Real Time reverse transcription—detection	ISO/TS 15216-2
*Yersinia enterocolitica*	Cultural examination—detection	ISO 10273
*Yersinia enterocolitica*	PCR Real Time—detection	ISO TS 18867

## Data Availability

Data is contained within the article or Appendix A.

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
