# Peer review of "Foodborne Toxigenic Agents Investigated in Central Italy: An Overview of a Three-Year Experience (2018–2020)"

_toxins, 2022, doi:10.3390/toxins14010040_

Round 1
Reviewer 1 Report
This report summarizes the incidence of foodborne pathogens and toxins found in food, environmental and clinical samples collected in relation to diagnosed or suspected FBD cases and submitted between 2018 and 2020 to the Food Microbiology Unit of the Istituto Zooprofilattico Sperimentale del Lazio e della Toscana. The topic is very important because food contamination and spoilage could be serious problems. I recommend that the manuscript is appropriate to be published in the journal as a report article.
Author Response
This report summarizes the incidence of foodborne pathogens and toxins found in food, environmental and clinical samples collected in relation to diagnosed or suspected FBD cases and submitted between 2018 and 2020 to the Food Microbiology Unit of the Istituto Zooprofilattico Sperimentale del Lazio e della Toscana. The topic is very important because food contamination and spoilage could be serious problems. I recommend that the manuscript is appropriate to be published in the journal as a report article.
Author response
We thank the reviewer. We agree with the suggestion and ask to the editor if it is possible to consider the publication of the article as a “report article”.
Reviewer 2 Report
The authors present a review on “Foodborne toxigenic agents investigated in central Italy: an overview of a three-year experience (2018-2020)”. The manuscript is clear, and the topic is interesting. However, Introduction and discussion should be revised. In the discussion authors should comment on their data; therefore, in my opinion, some of the information reported within the lines from 243 to 327 could be placed into the introduction. Instead in the introduction, the author could avoid describing in detail the different zoonotic agents.
In my opinion, if authors revise the instruction and the discussion the manuscript can be accepted for publication.
Author Response
The authors present a review on “Foodborne toxigenic agents investigated in central Italy: an overview of a three-year experience (2018-2020)”. The manuscript is clear, and the topic is interesting. However, Introduction and discussion should be revised. In the discussion authors should comment on their data; therefore, in my opinion, some of the information reported within the lines from 243 to 327 could be placed into the introduction. Instead in the introduction, the author could avoid describing in detail the different zoonotic agents.
In my opinion, if authors revise the instruction and the discussion the manuscript can be accepted for publication.
Author response
We thank the reviewer. Text has been modified according to the suggestion.